# Factors Associated with Cigarette, E-Cigarette, and Dual Use among South Korean Adolescents

**DOI:** 10.3390/healthcare9101252

**Published:** 2021-09-23

**Authors:** Myong Sun Cho

**Affiliations:** Department of Nursing, Kyongbok University, Kyonggi-do, Namyangju 12051, Korea; msunny.cho@gmail.com

**Keywords:** electronic cigarettes, adolescent health, tobacco use, dual use, electronic nicotine delivery systems

## Abstract

Dual use of e-cigarettes and cigarettes has become common among Korean adolescents but has decreased among adults. Dual use refers to using two tobacco products; however, in this study, it is defined as using both e-cigarettes and cigarettes. We assessed the prevalence of dual use among Korean adolescents and its relationship with socio-demographic, smoking-related characteristics, and other risk behaviors. The 2019 Korea Youth Risk Behavior Survey’s data on 57,303 adolescents were analyzed using logistic regression. Overall, 13.8% had recently (in the past 30 days), used some type of cigarette, 3.3% were dual users, 3.4% exclusively smoked conventional cigarettes, and 0.6% exclusively vaped e-cigarettes. After adjusting for socio-demographic and psychological factors, substance use, smoking initiation by 13 years, secondhand smoke in school and public, and amount of cigarette consumption proved significant for all cigarette user types. Dual use was strongly associated with younger age (≤15 years), cigarette smoking initiation before 13 years, secondhand smoke exposure at school, and heavy cigarette smoking. Limited smoking cessation attempts, secondhand smoke exposure in public, and ease of cigarette purchases decrease the odds of adolescents becoming dual users. Thus, surveillance and enforcement of the juvenile protection measures need updating to prevent a shift into dual use.

## 1. Introduction

Electronic nicotine delivery systems (ENDSs), of which electronic cigarettes are the most common prototype, deliver an aerosol by heating a solution that users inhale [1]. Therefore, in this study, ENDS and e-cigarette are used interchangeably. The e-cigarette industry has advertised that e-cigarette emissions are “water vapor”, which is why e-cigarette use (also known as “vaping”) is shown to be less harmful than traditional tobacco and occurs in public places. Similar to in the US [2] and the UK [3], after the introduction of e-cigarettes in Korea, their use increased, particularly among adolescents [4]. The current (previous 30 days) vaping rate and heat-not-burn tobacco smoking rate among 13- to 18-year-old Koreans are 3.2% and 2.6%, respectively; however, the smoking rate decreased from 13.3% in 2007 to 6.7% in 2019 [5]. In the US, the smoking rate for male adolescents was 9.3%, 2.5 times higher than that of female adolescents (3.8%), and smoking initiation occurred around the age of 13 years [6]. The same study found that a significant number of adolescents use both cigarettes and e-cigarette products, and 80.5% of current high school e-cigarette users concurrently smoke conventional cigarettes (dual use) [6,7]. In particular, teenagers are attracted to the variety of tastes, aromas, and sophisticated designs that e-cigarettes provide, however, those who start smoking with e-cigarettes become addicted to nicotine just the same [8]. In addition, some become dual users with the purpose of hiding their smoking from their teachers or parents at school or home [9]. Adolescent smokers may use e-cigarettes to aid in smoking cessation or reduce cigarette consumption [10].

Conflicting research results show that the odds of smoking cessation attempts significantly decreased in participants using e-cigarettes compared to cigarette smokers [11]. For this reason, a study by Lee and Ryu examined sex, urban environment, locality, fast food intake, the first smoking experience period, secondhand smoke (SHS) experience, alcohol consumption, and weekly allowance as risk factors for dual use [12]. According to a comparative analysis of e-cigarette and dual users, male and high school students having high stress levels, adolescents with alcohol experience, and participating in smoking prevention programs showed a higher probability of becoming dual users [13]. In addition, a study found that stress and feelings of hopelessness are associated with the use of cigarettes and e-cigarettes [14]; substance use was also positively correlated with cigarette use [15]. A previous study found significant interaction effects between the perceived accessibility of cigarettes and peer smoking [16] and SHS exposure [17]. Tobacco initiation at an earlier age is a risk factor for greater severity of nicotine dependence and difficulty in smoking cessation, and also increases the chances of adverse health consequences [18,19].

Several studies indicated positive associations between dual use and its different outcomes, including substance use [3,7,13,15]. However, a limited number of studies have examined the predictors of cigarette, e-cigarette, and dual lifetime use in samples of adolescents who were initially non-users [20]. Even fewer studies have examined the differences among current users of exclusive cigarettes, exclusive e-cigarettes, and dual users in terms of health risks [3]. Smoking-related patterns of exclusive cigarette users, exclusive e-cigarette users, and dual users can add to an emerging understanding of health risk behaviors. The health effects of e-cigarettes have not been well documented, and the long-term health risks potentially associated with vaping remain unknown. Therefore, with the advent of dual users, a study is needed to compare and review the comprehensive risk factors for these three types of users.

The present study examines the differences between four groups of 12–18-year-old adolescents (non-smokers, exclusive cigarette users, exclusive e-cigarette users, and dual users). The study aims to (1) estimate the prevalence of each user group and (2) identify the predictors of being in one of the three user groups compared to the non-smoker group. This study provides an understanding of socio-demographic factors, psychological factors, health risk behaviors, smoking-related factors, and different patterns of adolescent use of cigarettes and e-cigarettes. The results of our study may contribute to the development of youth smoking cessation programs and establishment of smoking cessation strategies through the analysis of factors affecting the three types of cigarette use.

## 2. Materials and Methods

### 2.1. Design and Subjects

This study presents a secondary data analysis of the 2019 Korea Youth Risk Behavior Survey (KYRBS). The KYRBS, conducted annually since 2005 by the Ministry of Education, Ministry of Health and Welfare, and the Korea Centers for Disease Control and Prevention (KCDC), provides national data on 15 categories of health risk behaviors associated with mortality, morbidity, and social problems using a 105-item questionnaire. 

A total of 800 middle and high schools were selected from a representative sample of 5661 schools, covering 2,683,547 adolescents aged 12 to 18, using the stratified multistage cluster sampling method. The 15th KYRBS was conducted with a final sample comprising 57,303 students (29,841 boys and 27,462 girls, mean age 15.08 ± 0.02 years), who agreed to participate in 2019. The questionnaire could be completed in approximately 45–50 min using computer-assisted self-reporting in school laboratories. The KYRBS was certified by Statistics Korea (Certificate Number: 117058) and approved by the institutional review board of the KCDC. Data were made available by the KCDC after permission was obtained on 15 June 2020 (http://www.kdca.go.kr/yhs/home.jsp) (accessed on 15 June 2020).

### 2.2. Measurements

Adolescent cigarette users were divided into three groups: (1) exclusive cigarette users (N = 1903)—students who used only cigarettes within the past month, (2) exclusive e-cigarette users (N = 335)—students who used only e-cigarettes within the past month, and (3) dual users (N = 1790)—students who used both cigarettes and e-cigarettes within the past month. The remainder of the sample were non-current users (N = 53,275).

The frequency of current (within the previous 30 days) use of cigarettes and e-cigarettes was assessed as follows: Group 1 was defined as those exclusively using cigarettes, the dependent variable, with the following question: “During the past 30 days, on how many days did you use cigarettes?”, similar to a previous study [2]. Group 2 was defined as those exclusively using e-cigarettes, the dependent variable, using the following question “During the past 30 days, on how many days did you use liquid e-cigarettes containing nicotine?” Group 3 included dual users of cigarettes and e-cigarettes for more than one day as the dependent variable. The response criteria were regrouped as follows: “0” for non-current users and “1 to every day” for cigarette users, e-cigarette users, and dual users.

According to the literature, factors influencing adolescents’ cigarette use are socio-demographic factors, psychological factors, health risk behaviors, and smoking-related factors. Therefore, 118 items from the KYRBS were extracted and used for analysis. Based on the literature review, independent variables such as socio-demographic factors (school type, residence type, sex, age, and perceived family economic status), psychological factors (perceived health status, stress, and feelings of hopelessness), and health risk behaviors (sexual intercourse, substance use, and treatment for violence-related injuries), were included. Residence type was classified as rural area, small or large city. Perceived family economic status was assessed using a 5-point (1: very low to 5: very high) Likert scale.

Smoking-related factors, including the initiation of cigarette smoking before age 13, smoking cessation attempts, SHS exposure at home, school, and public places, were surveyed as binary outcomes, and the participants answered either “Yes” or “No” to the questions.

The average number of cigarettes smoked in one day was assessed using the questions; “During the past 30 days, on average how many (cigarettes and e-cigarettes, respectively) did you smoke in a day?” and it was categorized as <1 cigarette, 1–9 cigarettes, 10–19 cigarettes, and ≥20 cigarettes.

The ease of cigarette purchase among participants was categorized as impossible to get, able to purchase cigarettes with effort, and easily purchased without any effort using the question, “How was it when you tried to purchase cigarettes from stores in the past 30 days?”

Perceived health status, stress levels, and SES were assessed using a five-point Likert-type scale (very good/high, good/high, moderate, bad/low, and very bad/low). Feelings of hopelessness were assessed using a yes or no question: “During the past 12 months, did you feel sadness or hopelessness?”

### 2.3. Statistical Analysis

To ensure unbiased national estimates, sampling weights were computed for the participants to ensure that the sample was representative of Korean students aged 12–18. A weighting factor was applied to each student record to adjust for probability of selection, non-response, and post-stratification adjustment to population estimates. The complex survey data were used to calculate weighted prevalence estimates of each tobacco use group.

The chi-square test was used to assess the relationship between types of cigarette use and socio-demographic, psychological, health risk, and smoking-related factors.

Multinomial logistic regression analysis was conducted, and non-users were compared to exclusive cigarette, e-cigarette, and dual user groups. Those models were adjusted for each factor, respectively. Statistical significance in this study was defined through *p*-values < 0.05, and adjusted *p* values with odds ratios (ORs) and 95% confidence intervals (CIs) were calculated using SPSS Statistics 27.0 (IBM Corp, Armonk, NY, USA).

## 3. Results

### 3.1. Socio-Demographic Factors

Of the 57,303 students who participated in the survey, 52.0% were male and 48.0% female. Most students resided in urban areas (94.4%), and students were relatively equally distributed between middle (47.9%) and high schools (52.1%; Table 1). Their average age was 15.08 ± 0.02 years, and SES was 2.64 ± 0.01.

Regarding the relationship between socio-demographics and cigarette use, male students had higher current rates of exclusive cigarette use (67.1%, *p* < 0.001), exclusive e-cigarette use (78.5%, *p* < 0.001), and dual use (78.1%, *p* = 0.871). High school students had significantly higher current rates of exclusive cigarette use (75.2%, *p* < 0.001), exclusive e-cigarette use (65.7%, *p* < 0.001), and dual use (79.6%, *p* < 0.001) than middle school students. Additionally, those older than 16 had significantly higher rates of current cigarette use compared to those aged 12–15 in terms of exclusive cigarette use (63.6%, *p* < 0.001) and exclusive e-cigarette use (51.2%, *p* < 0.001), as well as dual use (69.7%, *p* < 0.001; Table 1).

### 3.2. Psychological Factors

Exclusive cigarette (*p* < 0.001), exclusive e-cigarette (*p* < 0.001), and dual (*p* = 0.017) users showed significant differences in terms of levels of perceived stress (mean 2.71 ± 0.01). Perceived health status was significantly different between exclusive cigarette (*p* = 0.003) and exclusive e-cigarette (*p* < 0.001) users, not dual users (*p* = 0.641) (Table 1).

### 3.3. Health Risk Behaviors

Experience with substance use was significantly higher in exclusive cigarette users (2.2%, *p* < 0.001), exclusive e-cigarette users (17.8%, *p* < 0.001), and dual users (7.5%, *p* < 0.001) compared to non-current users (0.7%). Moreover, experience of treatment for violence-related injuries was significantly higher among exclusive cigarette users (4.9%, *p* < 0.001), e-cigarette only users (25.8%, *p* < 0.001), and dual users (11.2%, *p* < 0.001), compared to non-current users (1.9%) (Table 1).

### 3.4. Smoking-Related Factors

Of the 57,303 participants, 53,275 (92.7%) have not used either cigarettes or e-cigarettes within the previous month. A total of 1903 (3.4%) participants were exclusive cigarette users (male: 1255 vs. female: 648), and 335 (0.6%) were exclusive e-cigarette users (male: 259 vs. female: 76). Moreover, 1790 (3.3%; male: 1375 vs. female: 415) participants were dual users. In other words, these results show that dual use was five times more common than exclusive e-cigarette use.

Those who started smoking cigarettes before the age of 13 had significantly higher rates of being in the exclusive cigarette (34.4%, *p* = 0.092), exclusive e-cigarette (9.4%, *p* < 0.001), and dual use (53.4%, *p* < 0.001) groups than in the non-user group. Those smoking at least 20 cigarettes per day were more likely to be in the exclusive cigarette (3.9%, *p* = 0.092), exclusive e-cigarette (9.8%, *p* < 0.001), and dual use (12.7%, *p* < 0.001) groups than in the non-current user group.

Finally, students who were exposed to SHS at school had significantly higher rates of being in the exclusive cigarette (30.4%, *p* < 0.001), exclusive e-cigarette (44.8%, *p* < 0.001), and dual use (41.4%, *p* < 0.001) groups than in the non-current user group.

### 3.5. Factors Associated with Exclusive Cigarette Use

Being a male student increased the odds of exclusive cigarette use (OR, 2.455; CI, 1.732–3.487). Experience with substance use and treatment for violence-related injuries consistently increased the odds of exclusive cigarette use. In addition, initiation of cigarette smoking before the age of 13 years, SHS exposure at school, and average cigarette consumption per day were significantly associated with exclusive cigarette use compared to non-users.

Initiation of cigarette smoking before the age of 13 (OR, 3.872; CI, 2.526–5.935), secondhand smoke exposure at school (OR, 1.425; CI, 1.076–1.888), and increased amount of cigarette smoking (OR, 1.678; CI, 1.312–2.147) increased the odds of exclusive cigarette use.

In contrast, smoking cessation attempts (OR, 0.646; CI, 0.500–0.835) and SHS exposure at public places (OR, 0.709; CI, 0.52–0.965) reduced the odds of exclusive cigarette use. Among health risk behaviors, both substance use experience (OR, 5.420; CI, 3.162–9.292) and treatment for violence-related injuries (OR, 3.888; CI, 2.591–5.835) increased current cigarette use. Except for being male (OR, 1.20; CI, 1.12–1.29), none of the socio-demographic factors increased the odds of exclusive cigarette use. Additionally, only increased levels of perceived stress (OR, 1.224; CI, 1.044–1.435) from psychological factors increased the odds of exclusive cigarette use (Table 2).

### 3.6. Factors Associated with Exclusive E-Cigarette Use

Being a male student (OR, 1.666; CI, 1.367–2.031) increased the odds of exclusive e-cigarette use but higher family economic status (OR, 0.927; CI, 0.862–0.997) reduced the odds of exclusive e-cigarette use compared to non-users.

All health risk behaviors, sexual intercourse experience (OR, 1.317; CI, 1.126–1.540), substance use (OR, 2.035; CI, 1.358–3.050), and treatment for violence-related injuries (OR, 1.391; CI, 1.034–1.872) increased the odds of current e-cigarette use among smoking-related factors, and exposure to SHS at school (OR, 1.197; 95% CI, 1.034–1.387), exposure to SHS at public places (OR, 1.206; 95% CI, 1.031–1.410), and ease of cigarette purchase (OR, 1.179; 95% CI, 1.123–1.238) increased the odds of current e-cigarette use. Thus, the ease with which students could purchase e-cigarettes in stores and heightened SHS exposure at school and in public increased the odds of e-cigarette use.

Initiation of cigarette smoking before the age of 13 (OR, 0.707; 95% CI, 0.607–0.823) and an increased amount of cigarette smoking (OR, 0.650; 95% CI 0.581–0.729) reduced the odds of current e-cigarette use (Table 2).

### 3.7. Factors Associated with Dual Use

Older students (OR, 1.560; CI, 1.015–2.396) had increased odds of exclusive cigarette use compared to non-users. As for health risk behaviors, both substance use (OR, 3.182; CI, 1.925–5.258) and treatment for violence-related injuries (OR, 3.009; CI, 1.954–4.635) increased the odds of current dual use. In terms of socio-demographics, students aged 12–15 years (OR, 1.560; CI, 1.015–2.396) were more likely to use both cigarettes and e-cigarettes than those aged over 16 years. Among smoking-related factors, initiation of cigarette smoking before the age of 13 (OR, 5.569; 95% CI, 3.855–8.045), exposure to SHS at school (OR, 1.472; 95% CI, 1.047–2.070), and increased amount of cigarette smoking (OR, 2.492; 95% CI, 1.889–3.288) increased the odds of current dual use. Specifically, among smoking-related factors, SHS exposure at public places (OR, 0.517; CI, 0.367–0.729) and ease of cigarette purchase (OR, 0.888; CI, 0.798–0.990) reduced the odds of current dual use. However, none of the psychological factors reduced the odds of current dual use (Table 2).

## 4. Discussion

The prevalence rate results show that 92.7% of participants were non-current users (using neither cigarettes nor e-cigarettes), 3.4% were exclusive cigarette users, 0.6% were exclusive e-cigarette users, and 3.3% were dual users in the month prior to when the survey was conducted. The current dual use rate of 3.3% indicates that most exclusive e-cigarette users (0.6%) use both cigarettes and e-cigarettes. As a previous study showed, single use, such as exclusive cigarette use and exclusive e-cigarette use, decreased, and 28% of exclusive cigarette users converted to dual users after a 5-month follow-up of 371 single users [21].

US high school students’ current e-cigarette use increased from 11.7% to 20.8%, and middle school students’ use increased from 3.3% to 4.9% between 2017 and 2018. The rise in e-cigarette use among both middle and high school students has led to an increase in overall tobacco product use in the United States [2]. During the same period, the rate of e-cigarette smoking among Korean adolescents also increased from 2.2% to 2.7%. However, direct comparison with international data on adolescent e-cigarette use may not be appropriate due to the use of different survey methodologies, definitions of use, and age groups of participants.

In this study, dual use was not associated with stress, feeling of hopelessness, or sexual intercourse. However, there was a statistically significant association between stress and conventional cigarette use, and feelings of hopelessness and sexual intercourse were associated with e-cigarette use—this is consistent with the results of a previous study [2]. Since little is known about the association between dual use and the aforementioned factors, further research is required.

The current e-cigarette users had the highest relative proportion of substance use and treatment for violence-related injuries compared to the exclusive cigarette and dual user groups. This result is inconsistent with previous research showing that the dual use group consumed significantly more drugs than the exclusive e-cigarette group [20]. The current study’s results show that substance use is correlated with increased odds of exclusive cigarette use, exclusive e-cigarette use, and dual use.

The dual use of conventional cigarettes and e-cigarettes was strongly associated with the initiation of cigarette smoking before 13 years of age and an increased amount of cigarette smoking. Dual users might start smoking before the age of 13, and nicotine dependency becomes heavier with cigarette smoking [22]. One of the results of the current study indicates that the initiation of cigarette smoking before age 13 increases the odds of being a dual user by 5.6 times. This result is consistent with previous findings showing that smoking e-cigarettes and multiple other tobacco products before the age of 18 years is strongly associated with later daily cigarette smoking [23]. Preventing tobacco use or delaying the age at which adolescents start smoking can reduce the severity of nicotine dependence, as well as enhance the odds of successful smoking cessation [18].

Current e-cigarette use is strongly associated with exposure to SHS at school and public places and the accessibility of cigarettes. Consistent with previous studies [17], this study found that a higher frequency of SHS at school increased the odds of all types of cigarette use. Smoking is highly likely to be influenced by peers [24]. It was found that adolescents use e-cigarettes partly because of their curiosity and partly because their peers/friends do it. Adolescents are more likely to select friends whose smoking level is similar to their own, and peer influence is associated with smoking frequency [25].

In school, the number of adolescents who use e-cigarettes rather than conventional cigarettes has increased, and the use of e-cigarettes has increased as they experience SHS from friends’ use of e-cigarettes. For adolescents, e-cigarettes mimic and normalize smoking behavior, which can lead to nicotine addiction and, ultimately, to traditional tobacco consumption. Thus, it may be helpful to take a restrictive approach to advertising e-cigarettes and refilling containers [4].

This study reveals that adolescents are more likely to use e-cigarettes than cigarettes due to the former being easier to obtain. Currently, the purchase of conventional cigarettes is allowed from 18 years of age, and legal identification is required. If it is difficult for adolescents to directly purchase cigarettes due to these regulations, it is interpreted that they will instead choose e-cigarettes that can be purchased without identification procedures via the Internet. The closed system vaporizer (CSV) product, which has been on the Korean market since 2019, is small in size; the device itself does not look like a cigarette, has no smell and, unlike traditional cigarettes, no age restriction to purchasing it [26]. Therefore, adolescents can hide their smoking from adults and they choose the CSV as an alternative in situations where students may not smoke or need to hide the smell, which attracts particular attention. This makes it difficult for teachers and parents to detect their smoking behavior. The gap in regulations and surveillance increases the odds of adolescents becoming dual users. The introduction of these various e-cigarette devices in the market assists adolescents’ accessibility to cigarettes and increases the odds of e-cigarette use; thus, timely regulations should have accompanied its emergence in the market.

In this study, the low prevalence of e-cigarette use can be attributed to the dynamics of the accessibility and amount of tobacco consumption. E-cigarette users may evolve into dual users as their nicotine dependence grows, and the amount of tobacco consumption will increase after the initiation of e-cigarettes.

However, as this is a cross-sectional study, the temporal change in the type of user could not be verified; it needs to be examined through a further longitudinal study.

E-cigarette users indicated that a reason for use was that they could vape at times when, or in places where, smoking cigarettes was forbidden, and such use has consequently become common [27]. This secondary role of e-cigarettes may change with the more efficient e-cigarette nicotine delivery systems that have become popular in Korea [26].

Perceived accessibility also increases the risk of smoking progression among initiators in a dose–response fashion. The association between perceived accessibility and smoking is robust, particularly with respect to adopting peer and parental smoking habits. A previous study reported a positive association between tobacco accessibility and cigarette use or the dual use of cigarettes [28]. Tobacco-control policies that ban the sale of tobacco products to minors and limit smoking in public places are well known to reduce smoking behavior among adolescents. Thus, the Korean government needs to adopt evidence-based tobacco control policies, such as regulations and enforcement on illegal cigarette product vendors and the adoption of strong online age-verification procedures on the Internet market, to make it difficult for adolescents to purchase cigarettes [29].

Dual users have a significantly greater chance of initiating cigarette smoking before 13 years of age and an increased amount of cigarette smoking relative to those who only use either e-cigarettes or cigarettes. Dual use is strongly associated with a younger age (≤15 years), initiation of cigarette smoking before 13 years, fewer smoking cessation attempts, and heavier cigarette smoking. The perceived accessibility of cigarettes and SHS exposure in public places are likely to reduce adolescents’ dual use.

This study has certain limitations, including a cross-sectional design as well as self-report measures and a potential recall bias in asking students to remember how much they smoked and had been exposed to SHS in the past. Despite these limitations, our findings shed light on smoking-related patterns of exclusive cigarette, exclusive e-cigarette, and dual uses, thereby contributing to an increasing understanding of health risk behaviors.

## 5. Conclusions

This is the first study that highlights the growing wave of dual users of the current forms of tobacco consumption, in combination with ENDSs in students. The 2019 KYRBS is large and well-representative sample of the Korean students, therefore, the results may have external validity, which would be useful in making health decisions. In this study, the odds of dual use increased when smoking started before the age of 13, significant cigarette smoking was involved, and exposure to secondhand smoke at school occurred. Therefore, regardless of the type of cigarette, smoking cessation programs should delay the onset of smoking and prevent exposure to secondhand smoke through peers at school.

## Figures and Tables

**Table 1 healthcare-09-01252-t001:** Socio-demographic characteristics of participants (N = 57,303).

Factors		Total (N = 57,303)	Non-User (n, Weighted%) 53,275, 92.7%)	Current User (N = 4028)			*p*-Value
				Exclusive Cigarette User (n, Weighted%) (1903, 3.4%)	Exclusive E-Cigarette User (n, Weighted%) (335, 0.6%)	Dual User (n, Weighted%) (1790, 3.3%)	
Residence type	Rural	4497	4145 (5.5%)	166 (6.4%)	26 (5.5%)	160 (5.7%)	<0.001
	Urban	52,806	49,130 (94.5%)	1737 (93.6%)	309 (94.5%)	1630 (94.3%)	
School type	Middle school	29,384	28,350 (49.8%)	524 (24.8%)	115 (34.3%)	395 (20.4%)	<0.001
	High school	27,919	24,925 (50.2%)	1379 (75.2%)	220 (65.7%)	1395 (79.6%)	
Sex	Male	29,841	26,952 (50.3%)	1255 (67.1%)	259 (78.5%)	1375 (78.1%)	<0.001
	Female	27,462	26,323 (49.7%)	648 (32.9%)	76 (21.5%)	415 (21.9%)	
Age	≥16 years	23,161	20,631 (41.8%)	1165 (63.6%)	167 (51.2%)	1198 (69.7%)	<0.001
	≤15 years	33,908	32,499 (58.2%)	721 (36.4%)	156 (48.8%)	532 (30.3%)	
Perceived family economic status	Very low	1299	1073 (1.9%)	84 (3.9%)	22 (6.0%)	120 (6.2%)	<0.001
	Low	6042	5452 (10.1%)	294 (14.2%)	51 (16.4%)	245 (13.0%)	
	Middle	27,457	25,705 (48.0%)	892 (47.8%)	123 (37.6%)	737 (42.5%)	
	High	16,126	15,177 (28.9%)	454 (24.4%)	83 (24.4%)	412 (22.8%)	
	Very high	6379	5868 (11.1%)	179 (9.7%)	56 (15.6%)	276 (15.4%)	
Perceived health status	Very bad	322	251 (0.5%)	23 (1.3%)	8 (2.4%)	40 (2.2%)	<0.001
	Bad	3915	3579 (6.8%)	164 (8.6%)	28 (8.4%)	144 (7.9%)	
	Moderate	12,810	11,904 (22.6%)	443 (23.3%)	63 (17.9%)	400 (21.8%)	
	Good	24,785	23,292 (43.6%)	767 (41.0%)	116 (36.3%)	610 (34.8%)	
	Very good	15,471	14,249 (26.5%)	506 (25.8%)	120 (34.9%)	596 (33.3%)	
Stress	Very low	2235	2075 (3.8%)	44 (2.2%)	27 (7.4%)	89 (4.9%)	<0.001
	Low	8887	8356 (15.5%)	249 (13.4%)	57 (17.0%)	225 (12.8%)	
	Moderate	23,403	22,021 (41.5%)	661 (34.1%)	108 (33.8%)	613 (34.2%)	
	High	16,004	14,843 (28.1%)	584 (31.4%)	74 (21.8%)	503 (28.3%)	
	Very high	6774	5980 (11.2%)	365 (18.9%)	69 (20.1%)	360 (19.8%)	
Feeling of hopelessness	No	41,275	38,998 (73.0%)	1096 (58.5%)	195 (60.8%)	986 (55.5%)	<0.001
	Yes	16,028	14,277 (27.0%)	807 (41.5%)	140 (39.2%)	804 (44.5%)	
Sexual intercourse	No	54,021	51,350 (96.3%)	1405 (73.8%)	237 (69.6%)	1029 (57.9%)	<0.001
	Yes	3282	1925 (3.7%)	498 (26.2%)	98 (30.4%)	761 (42.1%)	
Substance use	No	56,700	52,911 (99.3%)	1858 (97.8%)	275 (82.2%)	1656 (92.5%)	<0.001
	Yes	603	364 (0.7%)	45 (2.2%)	60 (17.8%)	134 (7.5%)	
Treatment for violence-related injuries	No	55,902	52,260 (98.1%)	1808 (95.1%)	247 (74.2%)	1587 (88.8%)	<0.001
	Yes	1401	1015 (1.9%)	95 (4.9%)	88 (25.8%)	203 (11.2%)	
Initiation of cigarette smoking before 13	No	55,677					<0.001
	Yes	1594		632 (34.4%)	31 (9.4%)	931 (53.4%)	
Smoking cessation attempts	No	1279		580 (30.7%)	144 (41.7%)	555 (31.1%)	<0.001
	Yes	2749		1323 (69.3%)	191 (58.3%)	1235 (68.9%)	
SHS exposure at home	No	38,858	36,608 (69.2%)	1118 (59.6%)	164 (50.1%)	968 (54.4%)	<0.001
	Yes	18,445	16,667 (30.8%)	785 (40.4%)	171 (49.9%)	822 (45.6%)	
SHS exposure at school	No	45,270	42,698 (79.6%)	1333 (69.6%)	178 (55.2%)	1061 (58.6%)	<0.001
	Yes	12,033	10,577 (20.4%)	570 (30.4%)	157 (44.8%)	729 (41.4%)	
SHS exposure at public places	No	27,655	26,248 (48.5%)	730 (37.2%)	132 (40.0%)	545 (30.5%)	<0.001
	Yes	29,648	27,027 (51.5%)	1173 (62.8%)	203 (60.0%)	1245 (69.5%)	
Average number of cigarettes smoked per day	<1	54,223		551 (28.7%)	193 (57.6%)	204 (11.4%)	<0.001
	1–9	2196		1110 (58.6%)	93 (27.1%)	993 (55.8%)	
	10–19	541		163 (8.9%)	17 (5.4%)	361 (20.0%)	
	≥20	343		79 (3.9%)	32 (9.8%)	232 (12.7%)	
Ease of cigarette purchase	Easy	1268		25.6% (1071)	21.7% (193)	30.4% (587)	<0.001
	Difficult	1414		18.2% (486)	21.9% (71)	36.5% (557)	
	Impossible to get	54,621		25.6% (1071)	21.7% (193)	30.4% (587)	

SHS: Secondhand Smoke.

**Table 2 healthcare-09-01252-t002:** Factors associated with current cigarette, e-cigarette, and dual use.

Factor	Exclusive Cigarette Use			Exclusive E-Cigarette Use			Dual Use		
	*p*-Value	OR	95% CI	*p*-Value	OR	95% CI	*p*-Value	OR	95% CI
Residence type (Urban)	0.539	1.178	0.699–1.986	0.698	1.077	0.739–1.57	0.788	1.075	0.632–1.829
School type (High school)	0.509	1.161	0.745–1.808	0.465	0.896	0.668–1.202	0.486	1.174	0.747–1.844
Sex (Male)	<0.001	2.455	1.732–3.478	<0.001	1.666	1.367–2.031	0.597	1.097	0.778–1.547
Age ≥ 16 years Ref (Age ≤ 15)	0.058	1.457	0.987–2.15	0.464	0.907	0.698–1.178	0.042	1.560	1.015–2.396
Perceived family economic status	0.631	0.968	0.848–1.105	0.042	0.927	0.862–0.997	0.331	1.068	0.935–1.219
Perceived health status	0.942	0.994	0.848–1.166	0.139	0.942	0.87–1.020	0.942	0.994	0.850–1.163
Stress	0.013	1.224	1.044–1.435	0.263	1.047	0.966–1.135	0.071	1.147	0.988–1.332
Feeling of hopelessness	0.721	1.060	0.769–1.463	0.012	1.264	1.053–1.517	0.319	0.842	0.600–1.182
Sexual intercourse	0.148	1.250	0.924–1.691	0.001	1.317	1.126–1.54	0.344	0.861	0.631–1.175
Substance use	<0.001	5.420	3.162–9.292	0.001	2.035	1.358–3.050	<0.001	3.182	1.925–5.258
Treatment for violence-related injuries	<0.001	3.888	2.591–5.835	0.029	1.391	1.034–1.872	<0.001	3.009	1.954–4.635
Initiation of cigarette smoking before 13 years of age	<0.001	3.872	2.526–5.935	<0.001	0.707	0.607–0.823	<0.001	5.569	3.855–8.045
Smoking cessation attempts	0.001	0.646	0.500–0.835	0.383	1.072	0.917–1.253	0.001	0.621	0.475–0.812
SHS exposure at home	0.529	1.087	0.838–1.411	0.351	1.073	0.925–1.245	0.903	0.982	0.735–1.313
SHS exposure at school	0.014	1.425	1.076–1.888	0.016	1.197	1.034–1.387	0.026	1.472	1.047–2.07
SHS exposure at public places	0.029	0.709	0.52–0.965	0.019	1.206	1.031–1.410	<0.001	0.517	0.367–0.729
Average number of cigarettes smoked per day (Ref: <1)	<0.001	1.678	1.312–2.147	<0.001	0.65	0.581–0.729	<0.001	2.492	1.889–3.288
Ease of cigarette purchase (Ref: impossible to get)	0.186	1.068	0.969–1.178	<0.001	1.179	1.123–1.238	0.032	0.888	0.798–0.99

SHS: Secondhand Smoke.

## Data Availability

Data were made available by the KCDC after permission was obtained on 15 June 2020 (http://www.kdca.go.kr/yhs/home.jsp, accessed on 15 June 2020).

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
