# Peer review of "Factors Associated with Cigarette, E-Cigarette, and Dual Use among South Korean Adolescents"

_healthcare, 2021, doi:10.3390/healthcare9101252_

Round 1

Reviewer 1 Report

This article highlights the growing wave of users of the current forms of tobacco consumption, in a combination with new ones such as the e-cigarette in students. Among the main consumers are the new generations of adolescents, which have been addressed in this article, exploring factors that may generate new hypotheses for the use of this form of tobacco consumption. The methods are correctly described. The sample used in this article (15th Korea Youth Risk Behavior 78 Survey) is large and well representative of the group analyzed, so the results may have external validity, which would be useful in making health decisions. The analysis is simple but limited to the differences between the factor categories.

Some questions remain, which it would be important to clarify:

1) A 105-item questionnaire was applied to these subjects. However, it is not very clear why they were limited to the use of 18 factors analyzed.

2) There are only assessed differences between factors and groups independently, but it could show a better result to compare each group factor among cigarette preference group.

3) It is not clear if the average amount of cigarettes are daily or during a specific time period. If so, <1 cigarette it would be difficult to measure o may be better to categorize it as daily or occasional smokers.

4) Methods does not explain how e-cigarette quantities are measured. E-cigarettes are devices with a complex manner to quantify (e. g. per puffs or cartridge. doi: 10.1093/ntr/ntaa211)

5) There are many variables that are not referred at the Discussion section, as sex, stress, hopelessness, sex intercourse…)

Author Response

Thank you for giving me the opportunity to submit to Current Psychology a revised draft of my manuscript titled " Factors associated with cigarette, e-cigarette, and dual use among South Korean adolescents." I appreciate the time and effort that you and the reviewers have dedicated to providing your valuable feedback on my manuscript. I am grateful to the reviewers for their insightful comments on my paper. We have been able to incorporate changes to reflect most of the suggestions provided by the reviewers. I have highlighted the changes in the manuscript in red.

Below is a point-by-point response to the reviewers’ comments and concerns.

Comments from Reviewer 1

Comment 1: A 105-item questionnaire was applied to these subjects. However, it is not very clear why they were limited to the use of 18 factors analyzed.

Response: Thank you for this comment. To clarify the information presented, I have accordingly revised the manuscript (p. 3, line 4):

Factors influencing with adolescents’ cigarette use in literature such as socio-demographic factors, psychological factors, health risk behaviors, and smoking-related factors were influenced with cigarette use in previous studies were extracted from all 118 items from KYRBS and used for analysis.

Comment 2: There are only assessed differences between factors and groups independently, but it could show a better result to compare each group factor among cigarette preference group.

Response: Thank you for your suggestions. I analyzed again to present table 1 to compare each group factor among cigarette preference group.

Comment 3: It is not clear if the average amount of cigarettes are daily or during a specific time period. If so, <1 cigarette it would be difficult to measure o may be better to categorize it as daily or occasional smokers.

Response: Thank you for your suggestions. average amount of cigarettes was measured between “<1, 1, 2-5, 6-9, 10-19, 20” to the question “During the past 30 days, on average how many (cigarettes, e-cigarettes) did you smoke a day?”

As KYRBS measures the amount of cigarettes smoked, e-cigarettes, and dual use with one item, there is insufficient information on the amount of smoking for each group. Therefore, classifying daily or occasional smokers based on this is not attempted due to the risk of misclassification.

Comment 4: Methods does not explain how e-cigarette quantities are measured. E-cigarettes are devices with a complex manner to quantify (e. g. per puffs or cartridge. doi: 10.1093/ntr/ntaa211)

Response: Thank you for your suggestions.  To clarify further the e-cigarette quantities were measured from KYRBS. I have accordingly changed “The average amount of cigarette smoking in one day was assessed by the question; “During the past 30 days, on average how many (cigarettes, e-cigarettes) did you smoke a day? “and it was categorized as <1 cigarette, 1–9 cigarettes, 10–19 cigarettes, and ≥20 cigarettes.”

And unfortunately, we are not able to quantify e-cigarette use since no such data (per puffs or cartridge) provided from KYRBS.

Comment 5: There are many variables that are not referred at the Discussion section, as sex, stress, hopelessness, sex intercourse…)

Response: Thank you for your suggestions. Additional referred at the Discussion section, as stress, hopelessness, sex intercourse.

Reviewer 2 Report

Factors associated with cigarette, e-cigarette, and dual use 2 among South Korean adolescents

Review:

Thank you for the opportunity to review. This manuscript aims to explore factors related to cigarette use, e-cigarette use, and dual use of cigarettes and e-cigarettes. The author found unique factors associated with each outcome; however, the manuscript presents a lot of results, and it is somewhat difficult to follow what the key takeaways are. I believe that the manuscript can benefit from some additional clarity. My comments and suggestions are below for each section.

Abstract:

  • The first sentence reads as a sentence in a Methods section. I recommend rearranging the sentences so that the background comes first. Also, in the tobacco literature, dual use can refer to the use of two tobacco products not limited to e-cigarettes and cigarettes. You may try condensing the first two sentences into one, such as, “Dual use of e-cigarettes and cigarettes has become more common among Korean adolescents but has decreased among adults.” Then, define dual use specific to your study somewhere in the next sentence.
  • “Current” needs to be defined. Were the 3.4% who smoked only conventional cigarettes and 0.6% who vaped e-cigarettes also current users? I assume so, so please be explicit about this.
  • I advise using “vaped” instead of “smoked” when referring to e-cigarette use.
  • Change “young age” to “younger age”
  • Another language concern is the use of “likelihood” when referring to logistic regression results. I advise using “odds”. Please make this change throughout.

Introduction:

Overall, the Introduction could use some work. Based on the previously cited work, I am not sure how this study significantly contributes to the current literature. The section may benefit from some additional clarification on how this study is relevant to Korean youth.

  • P1, line 25: Do you mean “e-cigarette use (also known as “vaping”)”? And, to clarify, do all public places allow e-cigarette use in Korea?
  • P1, line 28: Do you mean “vaping”
  • P1, lines 29-32: What does “smoking” refer to? Cigarette smoking?
  • This sentence requires a reference: “In particular, teenagers are attracted to the variety of tastes, aromas, and sophisticated designs that e-cigarettes provide, but still, those who start smoking with e-cigarettes become addicted to nicotine just the same.”
  • P1, lines 39-40: Did the study that you referenced include adolescents? I am not sure if many adolescents use e-cigarettes as a smoking cessation tool. This is more common among adults. Please revisit the referenced article to clear this up.
  • P1, lines 42-44: You indicate that previous studies have examined these factors but only provide one reference.
  • Paragraph 3: This paragraph seems out of place. Perhaps combine it with the previous paragraph.
  • P2, line 68: Change “e-cigarettes-only users” to “e-cigarette-only users” (without the “s”)
  • P2, line 69: “Number” or “prevalence”? You are not just estimating the counts, right?
  • P2, lines 73-74: The language “frequency of smoking” is confusing. Your study examines frequency of use only for cigarettes.

Methods:

My biggest concern about the methods was the ways in which tobacco use groups were defined. Each tobacco use group was treated as a binary variable with 1 including those who indicated “yes” to using a product in the past 30 days and 0 include those who did not. The issue is that for each individual product outcome, other users may be in the denominator. For example, e-cigarette only users could have cigarette users in the denominator, thus comparing e-cigarette only users to non-users and cigarette users. This can be avoided by create a single nominal variable that includes non-users, e-cigarette only users, cigarette only users, and e-cigarette + cigarette dual users. Then, using multinomial logistic regression, each tobacco use group can be directly compared to non-users without issues of misclassification. Please consider adopting this method of analysis.

Design and Subjects

  • Please provide the year for the 15th Korea Youth Risk Behavior Survey
  • Most recent national surveys on tobacco use ask about multiple types of electronic nicotine products, commonly referred to as electronic nicotine delivery systems (ENDS). I highly recommend using this terminology if the KYRBS asks about more ENDS products in addition to traditional e-cigarettes.
  • P2, lines 92-95: Consider using the labels “exclusive” for students who used only cigarettes and only e-cigarettes, such as “exclusive cigarette users” and “exclusive e-cigarette users”, only if use groups are mutually exclusive as implied.
  • P3, line 96: Do non-users include former users too? Maybe use “non-current users”.
  • P3 should go below either in the Measurements section after the measures are described or in the first paragraph of the Results section.

Measurements

  • It is unclear how dual users were categorized. It is defined as adolescents who used cigarettes and e-cigarettes in the past 30 days, correct?
  • Please define the sociodemographic variables in this section such as how they were categorized.

Statistical Analysis

  • Please include a statement that you generated weighted prevalence of each tobacco use group overall and by each factor of interested in this section.
  • How many models were run? Three (one for each outcome)? This should be explained here.
  • Please indicate what models were adjusted for. I am assuming each model was adjusted for each factor in Table 2.

Results:

Only about 8% of the overall sample used cigarettes or e-cigarettes. Have you thought about analyzing this data among this subset alone as well? For example, comparing cigarette only users to e-cigarette only users?

  • Please consider reporting the overall weighted prevalence (n, %) of each tobacco use group in the Results and Table 1.
  • Please be consistent with your labels for each tobacco use category. For example, in the results, respondents are referred to as “cigarette-only users”, but in the Methods, they are labeled as “cigarette users”. These use groups need to be defined explicitly in the Methods.
  • For subheadings 3.5-3.7, I would refrain from using the term “influencing” as your study is cross-sectional, and it is inappropriate to imply temporality. I would use the phrase, “associated with” or something like this.
  • P6, line 182: Again, refrain from using the term “likelihood”. Your results demonstrate odds, not probabilities. Please make this edit throughout the manuscript.
  • P6, line 182: What does “any type of current cigarette use” mean? This language was used before in the paper, and I thought it was referring to cigarette vs. e-cigarette use. If it refers to any type of cigarette product, then I would just stick with “cigarette use”.
  • For results corresponding to the regression models, I recommend following a similar structure to table 1. That is, report results related to sociodemographic factors first, followed by health risk behaviors, etc.

Discussion:

Overall, this section is largely filled with the author repeating results and can benefit from some streamlining. What are your key takeaways? Those should be emphasized and fit in the context of existing literature. Also, how do your results fit within the context of tobacco regulatory science?

  • P8, lines 232-235: I am not sure how the last sentence is relevant to this study that is among Korean adolescents. Maybe include an additional sentence about use in Korea, or how estimates in your study compare to the US.
  • P8, lines 236-237: These proportions were statistically compared to non-users. If you want to compare across tobacco use groups, then you might need to run additional tests.
  • P8, lines 240-242: This is confusing. How does substance use increase the odds of dual use, but dual use does not increase the odds of substance use?
  • Paragraph 3 needs work. Earlier in the paper, it is mentioned that e-cigarettes can be used as smoking cessation attempts. Thus, it would make sense that there is no association between smoking cessation attempts and e-cigarette use if e-cigarette users are already using these products to attempt to quit. That could be one interpretation. Nevertheless, the proportion of youth who are attempting to quit is probably low, so I would be careful about how these results are interpreted.
  • P8, lines 253-254: This sentence requires a reference: “Dual users might start smoking before the age of 13, and nicotine dependency becomes heavier with cigarette smoking.”
  • P9, lines 274-292: The points made in this first paragraph contradict the points made in the following paragraph. It seems that e-cigarettes are easily obtainable in Korea, but you attribute the low prevalence of use to accessibility.
  • P9, lines 293-294: This paragraph belongs in the limitations.
  • P9-10, lines 308-311: Do you mean for e-cigarettes? I thought it was difficult for youth to already obtain cigarettes in Korea.
  • Please include a conclusory paragraph that restates your aims, highlights key findings, and concludes with recommendations for relevant stakeholders and health professionals.

Table 1:

  • It is unclear what tests the p-values correspond to, so please include a footnote. Do you compare each tobacco use group to non-users separately?
  • Small concern: The variables in table one should be presented in the order that they are presented in the Results section.

Table 2:

  • Please indicate the reference group for each exposure in this table. You did so for “average among of cigarette smoking” but none of the other factors.
  • Some odds ratios (95% CIs) are displayed to the hundredth decimal place, and some are displayed to the thousandth.

Author Response

Thank you for giving me the opportunity to submit to Current Psychology a revised draft of my manuscript titled " Factors associated with cigarette, e-cigarette, and dual use among South Korean adolescents." I appreciate the time and effort that you and the reviewers have dedicated to providing your valuable feedback on my manuscript. I am grateful to the reviewers for their insightful comments on my paper. We have been able to incorporate changes to reflect most of the suggestions provided by the reviewers. I have highlighted the changes in the manuscript in red.

Below is a point-by-point response to the reviewers’ comments and concerns.

Comments from Reviewer 2

Comment 1: I recommend rearranging the sentences so that the background comes first. Also, in the tobacco literature, dual use can refer to the use of two tobacco products not limited to e-cigarettes and cigarettes. You may try condensing the first two sentences into one, such as, “Dual use of e-cigarettes and cigarettes has become more common among Korean adolescents but has decreased among adults.” Then, define dual use specific to your study somewhere in the next sentence.

Response: Thank you for your suggestions. I agree with this comment.

I have changed accordingly as follows: Dual use of e-cigarettes and cigarettes has become more common among Korean adolescents but has decreased among adults. Dual use can refer to the use of two tobacco products and is defined as the use of both e-cigarettes and cigarettes in this study.

Comment 2: “Current” needs to be defined. Were the 3.4% who smoked only conventional cigarettes and 0.6% who vaped e-cigarettes also current users? I assume so, so please be explicit about this.

Response: I agree with this comment. I revised the text as follows:

“currently (in the past 30 days), 3.3% were dual users, 3.4% smoked exclusive conventional cigarette smokers, and 0.6% vaped exclusive e-cigarettes.”

Comment 3:  I advise using “vaped” instead of “smoked” when referring to e-cigarette use.

Change “young age” to “younger age”

Another language concern is the use of “likelihood” when referring to logistic regression results. I advise using “odds”. Please make this change throughout.

Response: Thank you for your suggestions. I agree with this comment.  I made changes accordingly “vaped e-cigarette”, “younger age” and “odd” from beginning to end.

Comment 4:  P1, line 25: Do you mean “e-cigarette use (also known as “vaping”)”? And, to clarify, do all public places allow e-cigarette use in Korea?

Response: Thank you for this comment. To clarify the information presented, I have accordingly revised the manuscript (p. 1, line 25): “(also known as “vaping”) is appeared to be less harmful than traditional tobacco use.”

The use of e-cigarettes is banned in public places, work places and public transport with the exception of designated smoking areas.

In Korea, the National Health Promotion Act regulates liquid e-cigarettes made to have the same effect as smoking by inhaling nicotine solution into the body through the respiratory system using an electronic device(ENDS). However, there is no regulation on Non-Nicotine Delivery Systems (ENNDS).

Comment 5:  P1, line 28: Do you mean “vaping”

P1, lines 29-32: What does “smoking” refer to? Cigarette smoking?

This sentence requires a reference: “In particular, teenagers are attracted to the variety of tastes, aromas, and sophisticated designs that e-cigarettes provide, but still, those who start smoking with e-cigarettes become addicted to nicotine just the same.”

Response: Thank you for this comment. I have made change “smoking”’ to refer to cigarette smoking and “vaping” to refer to e-cigarette smoking.

And I have provided a reference accordingly. Walley,S.C.; Wilson,K.M.; Winickoff,J.P.; Groner,J. A Public Health Crisis: Electronic Cigarettes, Vape, and JUUL

Comment 6:  P1, lines 39-40: Did the study that you referenced include adolescents? I am not sure if many adolescents use e-cigarettes as a smoking cessation tool. This is more common among adults. Please revisit the referenced article to clear this up.

Response: Thank you for this comment. I have made change referenced article included adolescents.

Comment 7:  P1, lines 42-44: You indicate that previous studies have examined these factors but only provide one reference.

Response: Thank you for this comment. I have made change “A study by Lee and Ryu examined  …”

Comment 8:  Paragraph 3: This paragraph seems out of place. Perhaps combine it with the previous paragraph.

Response: Thank you for pointing out these mistakes. I combine it with the previous paragraph. And this has been revised accordingly

Comment 9:  P2, line 68: Change “e-cigarettes-only users” to “e-cigarette-only users” (without the “s”)

Response: Thank you for pointing out these mistakes. I have made change “e-cigarette-only users”

Comment 10 :  P2, line 69: “Number” or “prevalence”? You are not just estimating the counts, right?

Response: Thank you for your suggestions. I have made change: “Number” to “prevalence”.

Comment 11 :  P2, lines 73-74: The language “frequency of smoking” is confusing. Your study examines frequency of use only for cigarettes.

Response: Thank you for this comment. This has been revised as follows (P2, lines 73-74):  

“The current work may contribute to the development of youth smoking cessation programs and establishment of smoking cessation strategies through analysis of factors affecting 3 type of cigarette use.”

Comment 12 :  defined. Each tobacco use group was treated as a binary variable with 1 including those who indicated “yes” to using a product in the past 30 days and 0 include those who did not. The issue is that for each individual product outcome, other users may be in the denominator. For example, e-cigarette only users could have cigarette users in the denominator, thus comparing e-cigarette only users to non-current users and cigarette users. This can be avoided by create a single nominal variable that includes non-current users, e-cigarette only users, cigarette only users, and e-cigarette + cigarette dual users. Then, using multinomial logistic regression, each tobacco use group can be directly compared to non-current users without issues of misclassification. Please consider adopting this method of analysis.

Response: According to the method suggested by the reviewer, each group was classified so that there was no overlap. And multinomial logistic regression method has been applied and revised.

Comment 13 :  Please provide the year for the 15th Korea Youth Risk Behavior Survey

Most recent national surveys on tobacco use ask about multiple types of electronic nicotine products, commonly referred to as electronic nicotine delivery systems (ENDS). I highly recommend using this terminology if the KYRBS asks about more ENDS products in addition to traditional e-cigarettes.

Response: I agree with this comment. I used 2019 KYRBS data, thus I made changes e-cigarettes to electronic nicotine delivery systems (ENDS) as follows

“Electronic nicotine delivery systems (ENDS), of which electronic cigarettes are the most common prototype, deliver an aerosol by heating a solution that users inhale (WHO. 2014).  Therefore, in this study, ENDS and e-cigarette are used interchangeably”.

Comment 14 :  P2, lines 92-95: Consider using the labels “exclusive” for students who used only cigarettes and only e-cigarettes, such as “exclusive cigarette users” and “exclusive e-cigarette users”, only if use groups are mutually exclusive as implied.

Response: I agree with this comment. This has been revised as follows (P2, lines 92-95): “exclusive cigarette users” and “exclusive e-cigarette users”

Comment 15:  P3, line 96: Do non-current users include former users too? Maybe use “non-current users”.

Response: I agree with this comment. This has been changed “non-current users” to “non-current users”.

Comment 16 :  It is unclear how dual users were categorized. It is defined as adolescents who used cigarettes and e-cigarettes in the past 30 days, correct?

Please define the sociodemographic variables in this section such as how they were categorized.

Response: Thank you for this comment. This has been revised as follows (P3, lines 105-106):  

 Group 3 included dual use of cigarettes and e-cigarettes use for more than one day as the dependent variable.

In response to your comment, I have written how to categorized sociodemographic variables as follows (p 3, lines 17–19): Residence was classified as rural area, small or large city. Perceived family economic status was assessed using 5- point (1: very low- 5: very high)  Likert.

dual users were categorized. More than one day both question. ‘During the past 30 days, how many days did you smoke cigarettes, even one cigarette?’ and ‘During the past 30 days, how many days did you use e-cigarettes?’, respectively.

ciga, e-ciga,

During the past 30 days, on average how many cigarettes did you smoke a day

<1, 1, 2-5, 6-9, 10-19, 20

Comment 17:  Please include a statement that you generated weighted prevalence of each tobacco use group overall and by each factor of interested in this section.

Response: Thank you for this comment. I have made the necessary corrections accordingly. Furthermore, I have rewritten the sentence as follows:

A weighting factor was applied to each student record to adjust for probability of selection, nonresponse, and post-stratification adjustment to population estimates. SPSS for statistical analysis of complex survey data, was used to calculate weighted prevalence estimates each tobacco use group.

Comment 18:  How many models were run? Three (one for each outcome)? This should be explained here.

Please indicate what models were adjusted for. I am assuming each model was adjusted for each factor in Table 2.

Response: Thank you for this comment. I have made the necessary corrections accordingly, combine with comment 12. Furthermore, I have rewritten the sentence as follows:

Multinomial logistic regression including all variables was used to assess the factors affecting exclusive cigarette use, exclusive e-cigarette use, and dual use. Those models were adjusted for each factor, respectively.

Comment 19:  Only about 8% of the overall sample used cigarettes or e-cigarettes. Have you thought about analyzing this data among this subset alone as well? For example, comparing cigarette only users to e-cigarette only users?

Response: Thank you for this comment. The current analysis is considered sufficient to identify factors influencing the use of various tobacco products, so further analysis is not performed.

Comment 20:  Please consider reporting the overall weighted prevalence (n, %) of each tobacco use group in the Results and Table 1.

Response: Thank you for this comment. I reported overall weighted prevalence (n, %)

Comment 21:  Please be consistent with your labels for each tobacco use category. For example, in the results, respondents are referred to as “cigarette-only users”, but in the Methods, they are labeled as “cigarette users”. These use groups need to be defined explicitly in the Methods.

Response: Thank you for this comment. In response to your comment 14,15, 21, labels for each tobacco use category have been revised as follows: exclusive cigarette users, exclusive e-cigarette users and non- current user

Comment 22: For subheadings 3.5-3.7, I would refrain from using the term “influencing” as your study is cross-sectional, and it is inappropriate to imply temporality. I would use the phrase, “associated with” or something like this.

Response: Thank you for this comment. Subheadings 3.5-3.7 has been changed “influencing” to “associated with”.

Comment 23: P6, line 182: Again, refrain from using the term “likelihood”. Your results demonstrate odds, not probabilities. Please make this edit throughout the manuscript.

Response: Thank you for this comment. This has been changed “likelihood” to “odds” throughout.

Comment 24: P6, line 182: What does “any type of current cigarette use” mean? This language was used before in the paper, and I thought it was referring to cigarette vs. e-cigarette use. If it refers to any type of cigarette product, then I would just stick with “cigarette use”.

Response: Thank you for this comment. I have revised “cigarette use”.

Comment 25:  For results corresponding to the regression models, I recommend following a similar structure to table 1. That is, report results related to sociodemographic factors first, followed by health risk behaviors, etc.

Response: Thank you for this comment. In response to your comment, I rewrite sociodemographic factors first and then health risk behaviors as follows: Being male student increased the odd of exclusive cigarette use (OR, 2.455; CI, 1.732–3.487). … Being male student (OR, 1.666; CI, 1.367–2.031) increased the odd of exclusive cigarette use but higher family economic status (OR, 0.927; CI, 0.862-0.997) reduced the odd exclusive cigarette use…. Older student (OR, 1.560; CI, 1.015-2.396) increased the odd of exclusive cigarette use.

Comment 26:  Overall, this section is largely filled with the author repeating results and can benefit from some streamlining. What are your key takeaways? Those should be emphasized and fit in the context of existing literature. Also, how do your results fit within the context of tobacco regulatory science?

Response: Thank you for this comment. I have rewrite conclusion

This is the first study highlights the growing wave of dual users of the current forms of tobacco consumption, in a combination with ENDS in students. The 15th KYRBS is large and well representative sample of the Korean students, so the results may have external validity, which would be useful in making health decisions.

Dual-use is a concerning behaviour among adolescents and one that is understudied. If e-cigarettes are being used in settings where smoking is prohibited, this may increase cumulative nicotine exposure and the associated adverse effects. It is possible that dual-use may represent a transition to tobacco product use, a process known as the gateway effect.

In this study, the odds of dual use increased when smoking started before age 13, smoking amount, and exposure to secondhand smoke at school. As previous study showed, single use, such as exclusive cigarette use and exclusive e-cigarette use, decreased, and 28% of exclusive cigarette users were converted to dual users after 5-month follow-up of 371 single users. Therefore, regardless of the type of cigarette, smoking cessation programs should delay the onset of smoking and prevent exposure to secondhand smoke through peers at school.

Comment 27:  P8, lines 232-235: I am not sure how the last sentence is relevant to this study that is among Korean adolescents. Maybe include an additional sentence about use in Korea, or how estimates in your study compare to the US.

Response: Thank you for this comment. In response to your comment, I made changes to the discuss section as follows (P8, lines 232-235):  During the same period, the rate of e-cigarette smoking among Korean adolescents al-so increased from 2.2% to 2.7%, a global trend.

Comment 28:  P8, lines 236-237: These proportions were statistically compared to non-current users. If you want to compare across tobacco use groups, then you might need to run additional tests.

Response: Thank you for this comment. compared to non-current users deleted

Comment 29:  P8, lines 240-242: This is confusing. How does substance use increase the odds of dual use, but dual use does not increase the odds of substance use?

Response: Thank you for this comment. I have revised accordingly.

dual tobacco use increased risk for alcohol, illicit drugs use.

Round 2

Reviewer 2 Report

Factors associated with cigarette, e-cigarette, and dual use 2 among South Korean adolescents

Review:

I appreciate the response to my previous review. I believe that the manuscript has been substantially enhanced through the first round of revisions and have only a handful more suggestions below.

Abstract:

  • “Smokers” after “smoked exclusive conventional cigarette” should be omitted to reduce redundancy.

Introduction:

The Introduction section has been greatly improved.

Methods:

Design and Subjects

  • I do not see many changes to the descriptive results, which must mean that the tobacco use categories were previously categorized as exclusive use and dual use. Is this correct?

Measurements

  • For the statement, “During the past 30 days, on average how many (cigarettes, e-cigarettes) did you smoke a day?” … To be clear, was this a separate question for cigarettes and e-cigarettes? Or was this one question describing both products?

Statistical Analysis

  • You mention using SPSS twice in this section.

Results:

  • For results from multinomial logistic regression models, it is helpful to include the referent group for the outcome in the text. This referent group also needs to be mentioned in the Statistical Analysis section. What group were you comparing the cigarette, e-cigarette, and dual users to? I am assuming non-users.
  • In section 3.7, you say: “Older student (OR, 1.560; CI, 1.015–2.396) increased the odds of exclusive cigarette use.” Instead, it would be that older students had increased odds of exclusive cigarette use. In other parts of the paper your language works because you are saying that “being x increased the odds of y.”

Discussion:

  • Line 267: Add “and” before sexual intercourse.
  • This passage: “However, there was a statistically significant proportional association between stress and conventional cigarette use, likewise feeling of hopelessness and sexual intercourse with e-cigarette use in consistence with previous studies [3,14]. Further study is required hence little is known about the association between dual use with the aforementioned factors” reads a little awkward to me. Also, the last sentence requires a period. I suggest omitting the word “proportional.”
  • Paragraph 5: This entire paragraph may not be as relevant given that the population of interest is youth and e-cigarettes are better smoking cessation tools for adults than they are for youth.

Conclusions

I like the addition of the conclusions section. However, this section is usually one paragraph long and includes the following: 1) Reiterating one or two key takeaways; 2) Public health implications including recommendations for policymakers or public health professionals

  • Reword the first sentence. It is wordy.
  • Gateway effect is not mentioned anywhere else in the paper. Either omit or discuss in the section above.

Table 2:

  • There is a “vs” in the one of the column headings that I do not believe belongs.

Author Response

Dear Editor and reviewer:

I wish to re-submit the manuscript titled “Factors associated with cigarette, e-cigarette, and dual use among South Korean adolescents.” The manuscript ID is healthcare-1280343.

We thank you and the reviewer for your thoughtful suggestions and insights. Our manuscript has benefited from these insightful recommendations. I look forward to working with you and the reviewer to move this manuscript closer to publication in the Healthcare.

We have reviewed the manuscript and made the necessary changes in accordance with the reviewer’s suggestions. The responses to all comments have been prepared and are attached herewith. In the revised manuscript, the main changes are reflected in red font.

The point-by-point response to the reviewer’s comments and concerns are given below.

Comment 1

Abstract: “Smokers” after “smoked exclusive conventional cigarette” should be omitted to reduce redundancy.

Response: Thank you for this comment. I have omitted “Smokers” after “smoked exclusive conventional cigarette.”

Comment 2

Methods:

Design and Subjects

I do not see many changes to the descriptive results, which must mean that the tobacco use categories were previously categorized as exclusive use and dual use. Is this correct?

 Response: Thank you for this comment. Yes, the descriptive results were previously categorized as exclusive use (cigarette and e-cigarette, respectively) and dual use.

Comment 3

Measurements

For the statement, “During the past 30 days, on average how many (cigarettes, e-cigarettes) did you smoke a day?” … To be clear, was this a separate question for cigarettes and e-cigarettes? Or was this one question describing both products?

Response: Thank you for this comment.  Yes, there were separate questions for cigarettes and e-cigarettes; to enhance clarity, I revised the text as follows:

“The average number of cigarettes smoked in one day was assessed using the questions; “During the past 30 days, on average how many (cigarettes and e-cigarettes, respectively) did you smoke in a day?“

Comment 4

Statistical Analysis

You mention using SPSS twice in this section.

Response: Thank you for bringing this to our/my attention Per your comment, I have omitted “SPSS for statistical analysis of.”

Comment 5

Results:

For results from multinomial logistic regression models, it is helpful to include the referent group for the outcome in the text. This referent group also needs to be mentioned in the Statistical Analysis section. What group were you comparing the cigarette, e-cigarette, and dual users to? I am assuming non-users.

Response: Thank you for this comment. Yes, non-users were compared to the cigarette, e-cigarette, and dual user groups.

I have made the following changes in the manuscript.

Results section

The referent group for the outcome is mentioned in the Results section.

Statistical Analysis section.

Multinomial logistic regression analysis was conducted, and non-users were compared to exclusive cigarette, e-cigarette, and dual user groups.

Comment 6

In section 3.7, you say: “Older student (OR, 1.560; CI, 1.015–2.396) increased the odds of exclusive cigarette use.” Instead, it would be that older students had increased odds of exclusive cigarette use. In other parts of the paper your language works because you are saying that “being x increased the odds of y.”

Response: Thank you for your suggestion. Accordingly, I have revised the sentence as follows.

Older students (OR, 1.560; CI, 1.015–2.396) had increased odds of exclusive cigarette use compared to non-users.

Comment 7

Discussion:

Line 267: Add “and” before sexual intercourse.

This passage: “However, there was a statistically significant proportional association between stress and conventional cigarette use, likewise feeling of hopelessness and sexual intercourse with e-cigarette use in consistence with previous studies [3,14]. Further study is required hence little is known about the association between dual use with the aforementioned factors” reads a little awkward to me. Also, the last sentence requires a period. I suggest omitting the word “proportional.”

Response: Thank you for your suggestions. Accordingly, I have omitted “proportional” and have revised the text as follows:

However, there was a statistically significant association between stress and conventional cigarette use, and feeling of hopelessness and sexual intercourse were associated with e-cigarette use—this is consistent with results of previous studies [2]. Since little is known about the association between dual use and the aforementioned factors, further research is required.

Comment 8

Paragraph 5: This entire paragraph may not be as relevant given that the population of interest is youth and e-cigarettes are better smoking cessation tools for adults than they are for youth.

Response: Thank you for your suggestion. I agree with this comment, and have, therefore, deleted Paragraph 5.

Comment 9

Conclusions

I like the addition of the conclusions section. However, this section is usually one paragraph long and includes the following: 1) Reiterating one or two key takeaways; 2) Public health implications including recommendations for policymakers or public health professionals

Reword the first sentence. It is wordy.

Gateway effect is not mentioned anywhere else in the paper. Either omit or discuss in the section above.

 Response: Thank you for your suggestions. Accordingly, I have revised the text as follows:

This is the first study that highlights the growing wave of dual users of the current forms of tobacco consumption, in a combination with ENDS in students. The 2019 KYRBS is large and well representative sample of the Korean students, therefore, the results may have external validity, which would be useful in making health decisions. In this study, the odds of dual use increased when smoking started before the age of 13, heavy/significant cigarette smoking was involved, and exposed to secondhand smoke at school. Therefore, regardless of the type of cigarette, smoking cessation programs should delay the onset of smoking and prevent exposure to secondhand smoke through peers at school.

Comment 10

Table 2:

There is a “vs” in the one of the column headings that I do not believe belongs.

Response: Thank you for bringing this to my attention. Per your comment, I have omitted “vs.”

Thank you for your consideration. I look forward to hearing from you.